# Molecular Pathways, Neural Circuits and Emerging Therapies for Self-Injurious Behaviour

**DOI:** 10.3390/ijms26051938

**Published:** 2025-02-24

**Authors:** Kristina Zhang, George M. Ibrahim, Flavia Venetucci Gouveia

**Affiliations:** 1Institute of Medical Science, University of Toronto, Toronto, ON M5S 3H2, Canada; 2Neurosciences and Mental Health, The Hospital for Sick Children, Toronto, ON M5G 0A4, Canada; 3Division of Neurosurgery, Hospital for Sick Children, Toronto, ON M5G 1X8, Canada

**Keywords:** self-injury behaviour, treatment, molecular basis, neuroanatomy

## Abstract

Nonsuicidal self-injurious behaviour (SIB) is a debilitating manifestation of physical aggression commonly observed across neurodevelopmental, psychiatric, and genetic disorders. This behaviour arises from a multifactorial aetiology involving genetic predispositions, epigenetic modifications, neurotransmitter dysregulation, and environmental stressors. Dysregulation in dopaminergic, serotonergic, glutamatergic, and GABAergic systems has been implicated in the pathophysiology of SIB, alongside structural and functional abnormalities within fronto-limbic-striatal circuits. These disruptions impair key processes, such as emotional regulation, reward processing, and behavioural inhibition, contributing to the emergence and reinforcement of SIB. Advances in preclinical research using genetic, lesion-based, pharmacological, and environmental animal models have been instrumental in elucidating the molecular and neurocircuitry underpinnings of SIB. Emerging neuromodulation therapies targeting critical nodes within the fronto-limbic-striatal network, particularly deep brain stimulation, have shown promise in treating severe, refractory SIB and improving quality of life. This review integrates current evidence from clinical studies, molecular research, and preclinical models to provide a comprehensive overview of the pathophysiology of SIB and therapeutic approaches. By focusing on the molecular mechanisms and neural circuits underlying SIB, we highlight the translational potential of emerging pharmacological and neuromodulatory therapies. A deeper understanding of these pathways will pave the way for precision-based interventions, bridging the gap between molecular research and clinical applications in SIB and related conditions.

## 1. Introduction

Self-injurious behaviour (SIB) refers to repetitive, self-directed actions that result in physical harm without suicidal intent [1]. Common forms include head banging, self-biting, excessive scratching, hair-pulling, and skin-picking, which can lead to serious injuries, medical complications, and a diminished quality of life for both individuals and caregivers [2,3,4]. These behaviours are observed across a wide range of neurodevelopmental, psychiatric, and genetic conditions, often complicating treatment and requiring intensive management [5,6].

SIB is highly prevalent in individuals with neurodevelopmental disorders, particularly autism spectrum disorder (ASD) [7], Fragile X syndrome [8], Lesch-Nyhan syndrome [9], Prader-Willi syndrome [10], Cri-du-Chat syndrome [11], and Cornelia de Lange syndrome [12]. Among individuals with intellectual disabilities of unknown aetiology, estimates of SIB prevalence range from 4% to 23% [13,14], but this rate increases significantly to 42–70% with a diagnosis of ASD [15]. The onset of SIB typically occurs in early childhood, with nearly half of the affected individuals developing these behaviours before the age of three and up to 90% by age ten [5,6,16]. Larger genetic studies indicate early onset patterns in specific syndromes [15]. In Fragile X syndrome, 12% of cases emerge by one year of age, 63% by four years, and 93% by eleven years. Similarly, in Prader-Willi syndrome and Down syndrome, SIB is present in 73–91% of cases by the age of seven [15].

While SIB manifests across diverse conditions, its characteristics differ depending on the underlying aetiology [17,18,19]. In ASD and in the context of intellectual disabilities, SIB is often stereotyped and repetitive, resembling other motor stereotypies such as hand flapping or body rocking [20,21]. In contrast, SIB in psychiatric disorders, including mood disorders, borderline personality disorder, and impulse-control disorders, is more commonly described as impulsive and emotionally driven, occurring in response to distress or dysregulation [5,16,22]. Some genetic conditions, such as Lesch–Nyhan syndrome, present with severe, compulsive self-injury, including self-biting and self-mutilation, which is strongly linked to neurobiological abnormalities in dopamine regulation [17,18,19]. These distinctions highlight the importance of precise characterisation when considering treatment approaches, as different mechanisms may drive SIB in different populations.

The motivations underlying SIB also vary, with some behaviours maintained by external social reinforcement and others occurring independently of environmental factors [16,23,24]. In socially maintained SIB, self-injury serves a function, which may include gaining attention from caregivers or escaping aversive tasks [16,23,24]. Individuals who struggle with communication may engage in SIB to signal distress or influence their environment [20,21]. By contrast, automatically reinforced SIB is self-sustaining and does not depend on external responses or vary with environmental stimuli. In this context, self-injury may regulate internal sensory experiences, reduce anxiety, or alleviate physiological discomfort [16,23,24]. This type of SIB is more commonly observed in individuals with ASD and profound intellectual disabilities, where it may emerge as a maladaptive strategy for managing sensory processing difficulties [20,21]. Some individuals exhibit a combination of social and automatic reinforcement, underscoring the complex, multifaceted nature of these behaviours [16,23,24]. These behaviours often persist into adulthood, causing severe physical harm and leading to profound negative impacts on quality of life [25]. Beyond the immediate physical injuries, SIB can result in social isolation, limited access to educational and vocational opportunities, and increased medical costs [25,26,27].

The origins of SIB cannot be attributed to a single factor but rather arise from an interaction between genetic, neurobiological, and environmental influences. Dysregulation of key neurotransmitter systems, particularly dopamine, serotonin, glutamate, and GABA, has been implicated in the development and persistence of self-injury [17,28,29]. Structural and functional abnormalities in cortico-striatal and limbic circuits, which regulate impulse control, reward processing, and emotional regulation, further contribute to the expression of SIB [30,31,32,33]. The overlap of these neural disruptions across multiple disorders suggests that common biological mechanisms may underlie diverse presentations of self-injury [17,28,29].

This review examines the molecular, neurochemical, and circuit-level mechanisms underlying SIB across neurodevelopmental, psychiatric, and genetic conditions. Drawing from genetic studies, neuroimaging research, and preclinical animal models, we explore how disruptions in neurotransmitter function and neural circuits contribute to self-injury. Additionally, we consider the functional and reinforcement mechanisms that sustain SIB, differentiating between socially mediated and automatically reinforced behaviours. Finally, we discuss current and emerging treatment strategies, including behavioural, pharmacological, and neuromodulatory interventions, focusing on translating neurobiological insights into therapeutic approaches. Understanding the interplay between molecular pathways and environmental influences is critical for developing targeted, individualised treatments that address the root causes of SIB and improve outcomes for affected individuals.

## 2. Genetic and Epigenetic Contributions

Genetic and epigenetic factors play a crucial role in the predisposition to SIB, particularly in individuals with neurodevelopmental and psychiatric disorders [18,19]. Genetic mutations, chromosomal anomalies, and epigenetic modifications contribute to the dysregulation of critical pathways involved in neurotransmitter function, synaptic architecture, and neural plasticity [18,19]. This section highlights the growing body of evidence from candidate gene studies, genome-wide association studies (GWAS), and epigenetic research, providing an overview of how genetic factors interact with environmental influences to drive SIB (Figure 1).

### 2.1. Genetic Mutations and Polymorphisms

Several genetic syndromes associated with intellectual and developmental disabilities include SIB as a core symptom, including Lesch-Nyhan, Fragile X, Prader-Willi, Smith-Magenis, Cri-du-Chat, Angelman, Lowe, and Cornelia de Lange syndromes [8,9,10,11,12,34]. In these syndromes, mutations in genes such as *HPRT1* (Lesch–Nyhan syndrome), *FMR1* (Fragile X), *UBE3A* (Angelman syndrome), *OCRL1* (Lowe syndrome), and the cohesin complex (Cornelia de Lange syndrome), as well as chromosomal deletions in 15q11-q13 (Prader–Willi and Angelman syndrome), 5p (Cri-du-Chat syndrome), and 17p (Smith-Magenis syndrome) disrupt critical pathways regulating neurotransmitter metabolism and synaptic function [17,35,36]. In ASD, mutations on *SHANK3* impair synaptic scaffolding and glutamatergic signalling, leading to aberrant cortico-striatal circuit activity, a hallmark of SIB [37]. Candidate gene studies have further identified polymorphisms in genes associated with neurotransmitter regulation as potential contributors to SIB. Variants in the serotonin transporter gene (*SLC6A4*), monoamine oxidase A gene (*MAOA*), and dopamine receptor genes (*DRD4* and *COMT*) have been linked to impulsivity, aggression, and repetitive behaviours [38,39,40]. For example, individuals with low-activity *MAOA* alleles exhibit heightened susceptibility to aggressive tendencies and SIB under stress [41,42]. Similarly, the brain-derived neurotrophic factor (BDNF) Val66Met polymorphism influences neurotrophic support and synaptic plasticity, modulating the risk for SIB in the presence of childhood trauma or stress [41,42,43].

### 2.2. Heritability and Genome-Wide Association Studies

Twin studies estimate the heritability of SIB to range from 30% to 70%, with higher rates observed in females [15,34]. These findings suggest that genetic factors significantly influence susceptibility to SIB, although shared environmental influences also play a role [15,34]. GWAS provide further insights, identifying loci associated with psychiatric traits that overlap with SIB, such as depression, ASD, and attention deficit hyperactivity disorder (ADHD) [44,45,46,47]. Notably, the netrin-1 receptor gene (*DCC*), which regulates prefrontal cortex (PFC) development, has been implicated in both suicidal and nonsuicidal self-injury, underscoring its relevance to circuit-level disruptions observed in SIB [48]. However, many GWAS studies have been limited by their inclusion of broad self-harm phenotypes, conflating nonsuicidal self-injury with suicidal behaviours [44,45,46,47]. This highlights the need for larger, more targeted studies to identify specific risk loci for SIB.

### 2.3. Environmental Interactions with Genetics and Epigenetics

The interaction between genetic predispositions and environmental factors plays a pivotal role in shaping the risk of SIB [18]. For instance, individuals carrying the short allele of the *SLC6A4* serotonin transporter gene exhibit increased vulnerability to SIB when exposed to severe interpersonal stress [38,49,50]. Similarly, carriers of the BDNF Val66Met polymorphism demonstrate greater susceptibility to the effects of childhood trauma, resulting in a heightened risk of self-directed aggression [43].

Epigenetic modifications represent a critical mechanism through which environmental factors influence gene expression, providing a link between external stressors and the biological pathways underlying SIB [51,52]. DNA methylation and histone modifications have been implicated in regulating genes involved in stress responses, neurotransmitter synthesis, and synaptic function [18]. For example, hypermethylation of the glucocorticoid receptor gene (*NR3C1*) has been observed in individuals with SIB and histories of childhood trauma, suggesting that dysregulated stress hormone signalling may mediate the link between adverse experiences and self-injurious behaviour [53]. Similarly, studies on methylation of the *SIRT1* promoter region in adolescents with depression and SIB indicate altered serotonergic transmission, highlighting the interplay between epigenetic regulation and neurotransmitter systems [54]. Furthermore, monoaminergic dysfunction observed in genetic syndromes is often exacerbated by external triggers such as sensory overload or social isolation [55,56,57].

Environmental factors, particularly early-life adversity, play a critical role in shaping the onset and expression of SIB. Impoverished institutional settings, chronic social isolation, and exposure to prolonged stress are strongly associated with increased SIB prevalence, emphasising the importance of environmental context in modulating the behaviour’s severity [58,59]. For instance, children raised in environments lacking social and sensory stimulation frequently exhibit repetitive and injurious behaviours, which may represent maladaptive responses to unmet needs for interaction and engagement [58,60]. In children with neurodevelopmental disorders, communication challenges further exacerbate the risk of SIB [61,62]. Social and environmental isolation often leads to reduced engagement with the surrounding environment, promoting SIB as a potential means of communication or social reinforcement [20,63]. Similarly, individuals with impaired communication skills or those in institutional settings for intellectual disabilities are particularly susceptible to SIB [64]. Additional environmental factors include physical discomfort, illness, or over-arousal from sensory stimuli, which may trigger or reinforce SIB [65]. Disruptions in the endogenous pain-opioid system and sensory reinforcement mechanisms have also been proposed as contributing factors [65]. These findings underscore the interaction between biological predispositions and environmental factors in determining the severity and persistence of SIB.

These interactions emphasise the importance of studying SIB within a biopsychosocial framework that accounts for the dynamic interplay between genetic susceptibility, epigenetic regulation, and environmental stressors. This integrative approach provides a more comprehensive understanding of SIB’s emergence, reinforcement, and persistence across diverse clinical populations. Despite these advances, the causal relationship between specific genetic variants and SIB remains elusive. Recent studies suggest that overlapping genetic and epigenetic variations across disorders may contribute to shared vulnerability for symptoms, such as impulsivity, aggression, and repetitive behaviours, which are core features of SIB [66,67]. Additionally, personal characteristics, including temperament, cognitive ability, and coping mechanisms, significantly modulate the severity and expression of SIB within and across syndromes [15]. Future research should prioritise large-scale GWAS and longitudinal studies to clarify the contributions of specific risk loci and gene-environment interactions. Incorporating multiomic approaches, such as epigenomics and transcriptomics, will also enhance our understanding of the molecular mechanisms underlying SIB and identify potential targets for intervention.

## 3. Neurobiological Basis of SIB

The development and expression of SIB are strongly linked to disruptions in neural circuits that mediate key processes, such as emotional regulation, reward anticipation, and behavioural inhibition [17,68,69,70]. Preclinical and clinical studies implicate specific neurotransmitter systems and structural abnormalities in fronto-limbic-striatal circuits as central to the pathophysiology of SIB [17,71]. This section explores how neurotransmitter dysregulation and structural circuit abnormalities contribute to SIB.

### 3.1. Neurotransmitter Dysregulation

Dysfunction in neurotransmitter systems is a key neurobiological feature of SIB [3,72,73]. The interplay between dopamine (DA), serotonin (5-HT), gamma-aminobutyric acid (GABA), and glutamate signalling underpins the regulation of behaviours, such as impulsivity, aggression, and repetitive motor actions, all of which are core features associated with SIB [74,75].

Dopaminergic insufficiency within the mesocorticolimbic and nigrostriatal pathways is one of the most consistently reported findings in SIB [76,77]. Dysregulation of dopaminergic receptor signalling has been consistently observed in individuals with SIB and animal models, contributing to an imbalance between motor activation and inhibition [17,71]. In particular, reduced DA signalling in the striatum contributes to impaired reward processing and a diminished capacity to suppress maladaptive behaviours [75]. Moreover, the hyperactivity of D1 receptor-mediated pathways (direct pathway) and hypoactivity of D2 receptor-mediated pathways (indirect pathway) contribute to an imbalance favouring impulsive and repetitive behaviours [15,17,78]. Pharmacological interventions that enhance dopaminergic signalling, such as the administration of levodopa in 6-hydroxydopamine (6-OHDA) lesioned rats, provide further insights into the pivotal role of DA in regulating SIB [79,80,81].

Serotonergic projections from the raphe nuclei modulate frontal-limbic interactions are critical for impulse regulation and emotional processing, and have been linked to impaired top-down inhibitory control [82,83]. Alterations in serotonin transporter (SERT) availability and receptor sensitivity (e.g., 5-HT1A and 5-HT2A) are implicated in heightened impulsivity and aggression, both precursors to SIB [84,85]. Evidence of rhesus monkeys subjected to social deprivation highlights the role of serotonin in mediating the effects of environmental stressors on SIB [78]. In the clinic, available pharmacological treatments for SIB will target the serotonergic system via selective serotonin reuptake inhibitors (SSRI) [2,86]. These findings emphasise the interaction between neurotransmitter dysregulation and environmental factors.

The balance between excitatory glutamatergic and inhibitory GABAergic signalling within cortico-striatal circuits is critical for behavioural regulation [17,87]. Imbalances in this system disrupt the excitation-inhibition equilibrium, leading to repetitive and injurious behaviours [77,88]. Reduced GABAergic tone, particularly within the PFC, amygdala, and striatum, has been associated with increased emotional reactivity and decreased capacity for behavioural inhibition, allowing maladaptive behaviours like SIB to emerge [17,87]. Excessive glutamatergic activity has been implicated in the hyperexcitability observed in SIB. Overactivation of NMDA (N-methyl-D-aspartate) receptors contributes to synaptic dysfunction and excitotoxicity, and antagonism of NMDA receptors exacerbates motor stereotypies and impulsivity [17,87,89]. Thus, the interplay between GABAergic and glutamatergic signalling is particularly critical within cortico-limbic-striatal circuits, where disruption in this balance leads to hyperactivation of excitatory pathways and a failure of inhibitory control, creating a permissive environment for the expression of repetitive and injurious behaviours [17,87,89].

Adenosine receptor subtypes (A1 and A2A) also modulate SIB through their interactions with dopaminergic and glutamatergic systems, further highlighting the interplay between multiple neurotransmitter systems in driving SIB [17,19]. The combined administration of A1 and A2A agonists in animal models reduces repetitive behaviours, highlighting the therapeutic potential of targeting this system [90].

### 3.2. Fronto-Limbic-Striatal Circuits

Structural and functional abnormalities within the fronto-limbic-striatal circuits are central to the neurobiology of SIB [17,19,82,91]. These interconnected networks regulate emotional responses, reward processing, and motor control, and their disruption contributes to the maladaptive behaviours observed in SIB [17,19,82,91] (Figure 2). Individuals with autism show accelerated striatal growth, particularly in the caudate nucleus, where the growth rate nearly doubles and is associated with more severe repetitive behaviours in childhood [92]. Studies show that SIB is related to reduced top-down inhibitory control of the frontal cortex over the limbic system, suggesting hyperactivation of the direct cortico-striatal-thalamo-cortical (CSTC) pathway and hypoactivation of the indirect pathway [91,93]. These aggressive behaviours are linked to amygdala and cingulate cortex hyperactivation [33]. These circuits mediate the balance between excitatory and inhibitory signals, essential for regulating motor behaviours, emotional responses, and reward processing [31,93].

The CSTC loop is a critical neural circuit that regulates motor behaviours, emotional responses, and impulse control, frequently disrupted in SIB [17,70,94]. The CSTC system operates through two interconnected pathways: the direct pathway, which facilitates motor output and goal-directed behaviours, and the indirect pathway, which suppresses inappropriate or excessive motor and emotional responses [17,70,94]. Dysregulation of these pathways underlies the neurobiological basis of SIB, with hyperactivation of the direct pathway and hypoactivation of the indirect pathway contributing to the observed maladaptive behaviours [17,19,95]. The direct pathway begins with excitatory input from the frontal cortex to the striatum, activating medium spiny neurons (MSNs) that express dopamine D1 receptors [19]. These neurons send inhibitory GABAergic signals directly to the globus pallidus interna (GPi) and the substantia nigra pars reticulata (SNr), which are major output nuclei of the basal ganglia. Under normal conditions, the GPi and SNr inhibit the thalamus, suppressing motor activity [19] (Figure 2). However, when the direct pathway is activated, its inhibitory GABAergic signals reduce this suppression, allowing the thalamus to send excitatory signals back to the frontal cortex [17,95]. This disinhibition promotes motor activity and facilitates the expression of goal-directed behaviours [17,95]. Dopaminergic input from the substantia nigra pars compacta (SNc) enhances the activity of D1 receptor-expressing MSNs, further amplifying the direct pathway’s effect, a process that is essential for initiating and sustaining motor actions and behavioural outputs [17,95].

In contrast, the indirect pathway suppresses excessive or inappropriate motor and emotional outputs [17,19,95]. In this pathway, cortical input activates striatal MSNs expressing dopamine D2 receptors, which project inhibitory signals to the globus pallidus externa (GPe) [17,19,95]. The GPe normally exerts an inhibitory influence on the subthalamic nucleus (STN). Inhibition of the GPe by the indirect pathway releases the STN from this suppression, allowing it to send excitatory glutamatergic input to the GPi and SNr [17,19,95]. This increased activity in the GPi and SNr enhances their inhibitory output to the thalamus, ultimately reducing motor activity [17,19,95] (Figure 2). Dopaminergic input from the SNc attenuates the activity of D2 receptor-expressing neurons, weakening the indirect pathway and favouring motor activation when required.

### 3.3. Neuroimaging Studies

Neuroimaging studies across species have identified consistent patterns of structural and functional abnormalities associated with SIB [30,32], revealing decreased striatal and thalamic volumes, altered patterns of neuronal activity of the amygdala and PFC, and fronto-limbic connectivity [30,32,90]. In children with ASD, Duerden et al. found that SIB was negatively correlated with the thickness of the right superior parietal lobule, bilateral primary somatosensory cortices, and volume of the left ventroposterior nucleus of the thalamus—key regions in the somatosensory system involved in sensory integration and body awareness [96]. Furthermore, structural magnetic resonance imaging (MRI) studies found the orbitofrontal cortex grey matter volume to be positively associated with the severity of restricted and repetitive behaviours in ASD [97]. Conversely, the right caudal anterior cingulate U-fibre volume was negatively associated with these behaviours [98]. Structural changes in subcortical regions, such as thalami, amygdala, and caudate nuclei, occurred in children with ASD and predicted the severity of repetitive restricted behaviours [99,100,101], further implicating these regions with impaired impulse control and emotional regulation.

Huang et al. reported that self-injurious thoughts and behaviours were associated with hyperactivation of the right amygdala, left hippocampus, and left posterior cingulate cortex—regions critical for emotional processing and mentalisation [33]. Functional MRI studies have further identified aberrant fronto-limbic activation, with over-activation of the PFC and nucleus accumbens (nAcc) alongside amygdala deactivation during pain stimulation in individuals with borderline personality disorder and SIB [99]. These findings align with other studies linking SIB to impulsivity, heightened stress reactivity, emotional dysregulation, and atypical pain sensitivity or modulation [63,65,102,103]. Notably, disruptions in regions such as the amygdala, anterior cingulate cortex, and basal ganglia highlight the interplay between pain perception, emotional processing, and the neural circuits underpinning SIB.

In animal models, such as the BTBR *T^+^ Itpr3^tf^*/J (BTBR) mouse model of SIB, decreased volume of the striatum and thalamus and increased volume of the hippocampus, cerebral cortex, and cerebellum have been associated with behavioural challenges [104]. Moreover, neuroimaging in C58/J mice showed that reduced volume in key cortical and basal ganglia regions, including the motor cortex, striatum, globus pallidus, and STN, was associated with repetitive behaviours [105]. Histological analyses of neonatal 6-OHDA-lesioned rodents and *Shank3*-deficient mice highlight similar neuroanatomical abnormalities, including striatal atrophy and disrupted corticostriatal synaptic connectivity [106,107]. These findings suggest a convergence of circuit-level dysfunctions across clinical and preclinical studies.

## 4. Animal Models of SIB

Animal models are critical in elucidating the molecular and circuit-level mechanisms underlying SIB and provide a platform for testing targeted interventions before clinical trials. These models provide invaluable insights into the genetic, neurochemical, and environmental factors contributing to SIB, enabling researchers to investigate its pathophysiology in controlled settings [29,108]. The strength of animal models for studying SIB is rooted in dimensions of face (the model’s ability to reflect clinical symptoms), construct (similar disease aetiology between the human condition and preclinical models), and predictive (response to treatments seen in clinical populations) validity [29,108]. Most animal models used in SIB studies demonstrate high face and construct validity and are thus valuable for investigating the pathophysiology and underlying mechanisms of potential treatments [29,108]. To date, numerous laboratory models have been generated in which developmental and neurochemical manipulations result in the expression of SIB. Since SIB can be a naturally occurring behaviour among animals in stress or deprivation environments, the manipulations applied to create laboratory models are thought to impact similar endogenous mechanisms underlying SIB in the more naturalistic contexts [29,108]. Animal models of SIB can be divided into four groups based on their induction method: (1) genetic models, (2) lesion models, (3) environmental manipulation, and (4) pharmacological manipulation [29,108]. Table 1 summarises commonly employed SIB animal models and key neurobiological findings from studies using these models.

### 4.1. Genetic Models

Genetically modified animals offer robust platforms for studying the molecular mechanisms of SIB. These models typically involve mutations or deletions of specific genes known to affect neural circuitry, neurotransmission, and behaviour [29,108]. Some well-characterised genetic models used to study SIB include those with gene knockouts in *Shank3*, *Fmr1*, *Slc6a3*, *Hoxb8*, and *Tsc1* or *Tsc2*, as well as those with mutations in *Mecp2* [29,108].

The *Shank3* knockout mouse has been developed to model the glutamatergic synaptic dysfunction and behavioural impairments of ASD [109,110]. These mice exhibit injurious self-grooming behaviours akin to SIB, resulting in lesions that typically first appear on the face or back of the neck [109]. Unlike other repetitive and restrictive behaviours, which are often nonharmful, this form of SIB specifically leads to self-inflicted physical injuries. A study using conditional knockout mice to investigate the effects of reduced *Shank3* expression in different brain areas revealed that while *Shank3* reduction in the striatum is associated with repetitive behaviours, *Shank3* deficiency in the hippocampus and cortex leads to injurious self-grooming [111]. Similarly, *Mecp2* mutant animals, which model Rett syndrome, display SIB and aggression linked to deficits in GABAergic and serotonergic systems [111,112,113]. Deleting *Mecp2* in GABAergic inhibitory neurons throughout the brain leads to injurious self-grooming, motor dysfunction, impaired social behaviour, and impaired memory [114]. However, if the loss of *Mecp2* occurs only in a subset of forebrain GABAergic neurons, no injurious self-grooming is observed [114]. Depletion of *Mecp2* in dopaminergic, noradrenergic, or serotonergic neurons induced motor impairments and aggressive behaviour [111,112,115,116]. The understanding of the *Mecp2* deficiency gained from these animal studies has led to the identification of a wide array of therapeutic targets, including the *Mecp2* gene and protein product, as well as the downstream mechanisms (i.e., neurotransmitter pathways, metabolic pathways, and ion channels) involved in SIB related to the *Mecp2* mutation [109,110].

Other genetic models, such as those with mutations in *Fmr1* (Fragile X syndrome model), *Slc6a3* (ASD model), *Hoxb8* (obsessive-compulsive disorder [OCD] model), and *Tsc1* or *Tsc2* (tuberous sclerosis model) similarly highlight disrupted monoamine signalling, which impacts impulse regulation and synaptic stability, both of which are key factors in SIB [117,118,119,120,121]. Similar to Fragile X patients, *Fmr1*^−/−^ animals do not produce FMRP, a protein involved in synaptic plasticity and function [121]. These animals have deficits in striatal GABA, glutamate, and 5-HT [117], which are associated with hyperactivity, social impairment, and cognitive deficits; however, their relationship concerning SIB is not yet clear [121,122]. Animals with DA transporter depletion, through the *Slc6a3* knockout mutation, exhibit high levels of extracellular DA, which is correlated with hyperactivity, impulsivity, and repetitive behaviour, including SIB [120]. This model is relevant for several fields of SIB research because of its clear and targeted dysfunction in DA regulation [120]. *Hoxb8* mutants exhibit increased cortical synapse and spine density within the frontal cortex along with increased dendritic spines in the dorso- and ventro-medial subregions of the striatum, suggesting that increases in excitatory corticostriatal synapses may be implicated in the elevated injurious self-grooming, anxiety, and social deficits in *Hoxb8^-/-^* animals [118]. Interestingly, long-term treatment of Hoxb8 mutants with the SSRI fluoxetine improves these behavioural impairments [118]. Finally, animals with *Tsc1* or *Tsc2* knockouts also exhibit repetitive behaviours, imbalanced excitation/inhibition, abnormal synaptic plasticity, and disruptions in the mTOR signalling pathway—a crucial regulator of cell growth and synaptic function [119]. The degree of behavioural and cognitive deficits depends on the specific *Tsc1* or *Tsc2* mutation, underscoring the complexity of SIB in the context of neurodevelopmental disorders [119]. This variability highlights the need for further research to refine these models and enhance their translatability to the clinical population.

Although not produced through direct genetic manipulation, the inbred BTBR mouse is among the common model systems used for studying SIB in the context of ASD [123,124]. Similar to the previously mentioned rodent models, this strain exhibits excessive levels of self-grooming and repetitive behaviour [123,124,125,126], along with several neuroanatomical abnormalities analogous to those observed in patients with ASD and co-morbid SIB [104]. BTBR mice exhibit altered DA, 5-HT, GABA, glutamate, and noradrenaline systems. Notably, reduced DA levels in the amygdala [127], accompanied by increased DA metabolites [127] and compromised DA D2-mediated neurotransmission [128], are associated with excessive self-grooming episodes. The animals also exhibit substantially blunted striatal and mesolimbic DA neurotransmission, an effect related to dysfunctional pre- and postsynaptic D2R signalling, with a plausible contribution of striatal adenosine A2A alterations [128]. An imbalance in excitatory and inhibitory neurotransmission has also been observed, characterised by reduced GABA signalling, with lower GABA levels detected in the amygdala [127], PFC, and hippocampus [129], alongside elevated glutamate levels in the amygdala [127]. Studies of 5-HT in BTBR mice revealed reduced 5-HT transporter (SERT), increased 5-HT1A receptor signalling capacity, and altered behavioural responses to 5-HT1A receptor ligands [130,131]. Interestingly, the excessive, injurious self-grooming expressed by BTBR mice is blocked by mGluR5 antagonist MPEP [124], 5-HT receptor antagonists [132,133], the adenosine 2A agonist CGS2168 [134], muscarinic cholinergic receptor agonist oxotremorine [135], without affecting locomotion. Taken together, the inbred BTBR mouse strain shows dysregulation across multiple monoamine systems essential for controlling impulsivity and reward processing. Dysfunction in these systems appears to drive excessive, injurious self-grooming behaviour observed in these animals, making BTBR mice a valuable model for studying SIB.

### 4.2. Lesion Models

Lesion models, such as neonatal 6-OHDA lesioned rodents, replicate dopaminergic deficits observed in conditions such as Lesch-Nyhan syndrome [29,108,136,137]. This model is among the most well-characterised lesion models for studying SIB, in which dopaminergic neurons of the striatum are destroyed in neonatal rodents [29,108,136,137]. However, it is important to note that the 6-OHDA model is only effective for modelling SIB if the dopaminergic neurons are almost completely destroyed during the neonatal stage, as the behavioural manifestations of lesioning are age-dependent [137]. Early studies have shown that neonatal rats with reduced DA and NE remained indistinguishable from nonlesioned controls, except for a slight decrease in body weight [138] and enhanced levels of stereotypic self-grooming [139]. When exposed to dopamine agonists (e.g., levodopa, apomorphine) in adulthood, 6-OHDA lesioned rodents exhibit immediate and profound SIB (i.e., self-biting, self-grooming) [79,80,81], demonstrating the critical role of dopaminergic signalling in the expression of self-directed behaviours.

The 6-OHDA lesion model exhibits decreased binding to D1R but increased binding to D2R, thus implying that DA-agonist-induced SIB may result from actions through D2R [79,80,81]. However, studies have contradicted this notion whereby administration of the D1R agonist SKF 38393 has been shown to cause greater inhibitory responsiveness of spontaneously firing neostriatal units in 6-OHDA rats [140]. In contrast, responses to the D2R agonist PPHT were relatively unaffected [140]. In addition, administration of D2R agonists induces hyperlocomotion and stereotypy, but no SIB in neonatally lesioned rats [81,141,142]. These findings highlight that although 6-OHDA lesions do not increase the expression or binding of D1R, the increased sensitivity of signalling through this class of receptors is crucial in SIB expression among neonatally lesioned rat models.

Importantly, lesion models have revealed the dynamic interplay between DA and other neurotransmitter systems, such as serotonin, GABA, and noradrenaline, in modulating SIB. Adult rats lesioned during the neonatal stage exhibit increased GABA, substance P, and met-enkephalin [81,143]. These changes in GABA and neuropeptide concentration suggest that DA depletion during neonatal development may cause functional changes to the medium spiny neurons of the striatum since GABAergic and peptidergic neurons are principal targets of dopaminergic neurons [81,143]. Animals with neonatal 6-OHDA lesions also show increased 5-HT innervation in the striatum during adulthood [144]. This heightened innervation is associated with enhanced binding of 5-HT_1B_ and 5-HT_2_ receptors, along with increased sensitivity to 5-HT receptor agonists [144]. Given the involvement of DA on SIB expression in the neonatal 6-OHDA lesion model, one might expect 5-HT also to play a crucial role. However, the findings are inconsistent: some studies indicate that 5-HT administration does not induce SIB, while others show that systemic administration of 5-HT antagonists also does not affect SIB [29,108,136,137]. Analysis of the potential contributions of other monoaminergic systems reveals that the noradrenergic system is likely not involved in SIB expression [80]. Breese et al. found that inhibition of DA-β-hydroxylase, an enzyme that converts DA to NE in noradrenergic neurons, does not affect SIB expression in the neonatal 6-OHDA model [80]. These insights have provided a plausible rationale for the utility of pharmacological interventions targeting monoaminergic pathways.

### 4.3. Early Environmental Deprivation

Models of early-life environmental deprivation and social isolation provide valuable insights into the impact of environmental factors on SIB and are particularly useful for studying the interaction between genetic predispositions and environmental stressors in shaping behaviour [29,108,137]. Neglect and abuse in childhood are thought to contribute to psychiatric disorders in which SIB may be a common feature [58,61,64,145,146,147]. Thus, the important etiological similarities of this model and the development of SIB in human psychopathologies underscore the construct validity of these early environmental deprivation models used within the laboratory.

These models originated from observations that nonhuman primates demonstrate aberrant behaviours, including SIB, when raised in stressful or socially impoverished environments [148,149,150]. In rhesus monkeys, this pathology usually takes the form of excessive hair-pulling, head-banging, and self-directed biting that can lead to severe tissue damage and mutilation [150,151,152]. Analyses of colony and veterinary records show that several risk factors for developing SIB in macaques are related to the isolating and captive nature of individual cage housing. Similar to the clinical population, in nonhuman primates, the age of the first individual housing [153,154], the proportion of the first 48 months of life spent in solitary caging [155], and the total duration of individual housing [153,155] significantly predict the development of self-inflicted wounding later in life. Interestingly, males are more likely to develop SIB than females [153,154].

Stress also plays a significant role in SIB expression among early environmental deprivation models. Isolation-reared animals often express SIB in the context of environmental and emotional stress [149,151], similar to that observed in humans with SIB [4,156]. Rhesus monkeys raised in socially stressful conditions and exposed to acute stress demonstrate elevated levels of emotional dysregulation, anxiety, fear, and aggression [157] and exhibit altered functioning in the hypothalamic-pituitary-adrenocortical axis, including blunted cortisol response, which is associated with increased levels of SIB [152,158].

Furthermore, maternally deprived and isolation-reared rhesus macaques exhibit lower DA metabolite (3,4-dihydroxyphenylacetic acid) concentration and apomorphine (D2 agonist) sensitivity, respectively, suggesting that early environmental deprivation can permanently change DA receptor sensitivity [159,160]. Altered 5-HT transmission is also implicated in the expression of SIB in rhesus monkeys, where treatment with the 5-HT precursor 1-tryptophan resulted in significant reductions in self-directed biting in 7 rhesus monkeys with a history of self-directed wounding [161]. These findings suggest that high levels of self-biting in monkeys could be due to a deficiency in 5-HT neurotransmission [161]. Moreover, rhesus macaques exposed to early environmental deprivation exhibit pronounced loss of striatal patch/matrix organization and chemoarchitecture in adulthood [162].

Rodent models of environmental deprivation similarly demonstrate increased dopaminergic activity and impaired inhibitory control, mirroring the neurobiological disruptions seen in human SIB [163,164]. Typically, for rodents, environmental deprivation can be presented as prolonged social isolation or a lack of environmental enrichment [163,164]. In fact, the addition of environmental enrichments in the housing cage, such as nesting material, reduced excessive self-grooming behaviour in rats [164]. Rats exhibiting environmentally induced SIB have higher basal concentrations of extracellular DA in the nAcc [165] and striatal area [165,166], as well as elevated basal levels of D1R binding in the caudate and lower levels of D2R binding in the caudate and nAcc [167].

Taken together, the findings in these animal models suggest that SIB induced by early environmental deprivation may be related to altered DA and 5-HT neurotransmission and result in hypersensitivity to DA agonists. Nevertheless, it is important to note that environmentally induced models present several challenges for studying SIB due to the inherent heterogeneity of animals (i.e., genetic factors, personality traits, previous experiences, etc.), which can impact the model’s vulnerability to SIB and response to treatment.

### 4.4. Pharmacologic Models

Pharmacological models, such as those using pemoline [168,169,170,171], methamphetamine [172], or GBR-12909 [173], induce SIB through targeted disruption of neurotransmitter systems. These models highlight the role of several neurotransmitter systems in the pathophysiology of SIB, particularly monoamine imbalances in driving self-directed behaviours [108,137,168,169,171,174]. SIB may also be modelled by the administration of caffeine [171,175], the adrenergic agonist clonidine [176], or the calcium channel agonist Bay K 8644 [177,178].

Pemoline-induced SIB in rodents has provided strong predictive validity for pharmacological treatments, offering insights into the efficacy of dopaminergic and serotonergic agents [168,169,171,174]. In this model, high doses of pemoline, a long-lasting indirect monoamine agonist that blocks DA, 5-HT, and NE reuptake, are administered to rats to induce SIB [174]. This model is particularly interesting because patients with amphetamine-induced psychosis often exhibit severe SIB [179,180]. Additionally, repeated pemoline treatment results in approximately 30% depletion of striatal DA, resembling the degree of DA loss observed in Lesch–Nyhan syndrome [108] and further supporting the notion that drug-induced alterations to DA mechanisms play an important role in the aetiology of stimulant-induced SIB.

Pemoline also reverses DA transporter functioning, causing a presynaptic release of DA while simultaneously blocking its reuptake, and can have its effect reverted by administration of DA antagonists [181]. One particularly interesting study by Cromwell et al. compared the responses of striatal MSNs to cortical stimulation in brain sections from pemoline-treated rats that exhibited SIB to those that did not [182]. They found that DA application increased the evoked depolarizing potential responses of neurons in pemoline-treated SIB rats when compared to non-SIB and control rats [182]. Moreover, this response to DA in neurons of SIB rats required co-activation of D1R and D2R and was blocked by NMDA receptor antagonist 2-amino-5-phosphonovaleric acid. Glutamatergic and dopaminergic neurotransmission are highly interactive in the CSCT loop, and administration of the NMDA receptor antagonist MK-801 before pemoline blocks SIB in rats [183,184], an effect not observed in rats given MK-801 after pemoline injection [184]. Overall, these findings suggest that glutamate-induced neuroplasticity, along with altered cortico-striatal glutamatergic and dopaminergic signalling in MSNs, may contribute to the pathophysiology of pemoline-induced SIB.

The calcium channel agonist, Bay K 8644, induces dystonia and SIB in mice, especially if administered during early post-weaning development. Studies have found that SIB is augmented by the administration of the indirect DA agonists amphetamine and GBR 12909 [177], the monoamine oxidase inhibitor clorgyline [185], and the SSRI fluoxetine in this specific model [185]. Furthermore, Bay K 8644-induced SIB was suppressed when 5-HT or vesicular stores of DA were depleted [185], indicating the involvement of 5-HT and DA systems in SIB manifestation in this model. Bay K 8644-induced SIB was also attenuated by co-administration of D1/D5 receptor antagonists (i.e., SCH-23390, SKF-38566), as well as D3 antagonists (i.e., U-99194, GR-103691) [186]. In contrast, co-administration of D2 (i.e., L-741,626) and D4 (i.e., L-745,870) antagonists did not affect Bay K 8644-induced SIB. The effects of Bay K 8644 were also weakened in D3 receptor knockout mice while being amplified in D1 knockouts. These findings suggest that DA and 5-HT neurotransmission play important roles in SIB induction by Bay K 8644 in mice, and D1, D3, and D5 receptor signalling processes mediate the SIB onset.

An intriguing model previously explored for its perceived reliability in simulating SIB is the chronic caffeine model [171,175]. This approach was initially promising, as caffeine is thought to influence DA functioning via the antagonistic effects of methylxanthine on adenosine receptors, modulating presynaptic DA neurotransmission and altering postsynaptic DA responses [171,175]. However, more recent studies show that only a small proportion of caffeine-treated animals display SIB, which, when present, is typically mild [171]. Additionally, many caffeine-treated animals experience adverse effects, including weight loss, chromodacryorrhea (i.e., red lacrimal secretion), thymus involution, and even death [171]. Thus, these severe side effects make caffeine-induced self-injury an unreliable model, limiting the potential for meaningful biochemical or behavioural analysis [29,171].

Future research should refine existing animal models to enhance their translational relevance. Incorporating advanced techniques such as optogenetics, chemogenetics, and in vivo imaging will enable researchers to dissect the molecular and circuit-level dynamics of SIB with greater precision. By integrating findings from genetic, lesion, and pharmacological models, researchers can develop a comprehensive understanding of SIB and its molecular underpinnings, paving the way for innovative treatments.

## 5. Treatment Approaches for SIB

The neurobiological basis of SIB reflects complex interactions between neurotransmitter dysregulation, structural abnormalities in fronto-limbic-striatal circuits, and environmental influences. This multifactorial aetiology necessitates an integrated, multidisciplinary approach to treatment that targets the underlying molecular, neurocircuitry, and psychosocial mechanisms of SIB [68,86,187,188]. Current strategies include behavioural interventions, pharmacological treatments, neuromodulation techniques, and emerging molecular-targeted therapies [86,189]. Table 2 provides a summary of the main therapeutic strategies currently employed for the clinical management of SIB. Although significant progress has been made, there remains an unmet need for evidence-based, individualised treatment approaches to manage severe SIB effectively.

### 5.1. Behavioural Interventions

Behavioural interventions form the cornerstone of SIB management, particularly for individuals with neurodevelopmental or psychiatric disorders [86,187,190], as has been designated as the “best practice” for children and adolescents with SIB by the American Academy of Paediatrics [189]. These treatment and prevention strategies employ function–based and patient-specific treatment programs to identify proximal stressors that may elevate the risk of SIB and develop skills to cope with such stressors more effectively [189,191,192]. These approaches are best implemented in a family-based setting, where caregivers are trained to identify triggers and implement consistent behavioural strategies that mitigate SIB [86,187,190]. While physical restraints (e.g., gloves, helmets) may be necessary to prevent serious injury during acute episodes, their use should remain temporary and closely monitored [86,187,190]. Interventions should prioritise teaching functional communication skills and adaptive coping mechanisms to reduce reliance on restrictive measures [86,187,190].

Behavioural strategies, including functional behaviour assessments (FBA) and applied behaviour analysis (ABA), have demonstrated significant efficacy in identifying triggers and developing targeted interventions to reduce SIB [86,187,190]. FBA systematically assesses the functional role of SIB (e.g., sensory stimulation, attention-seeking, or escape from aversive situations) and informs the implementation of individualised treatment plans. For example, if SIB serves as a mechanism for avoiding difficult tasks, behavioural interventions focus on teaching alternative communication strategies, such as verbal requests for breaks or support [86,187,190]. ABA uses reinforcement techniques to encourage adaptive behaviours while reducing maladaptive ones [86,187,190]. These include contingency management strategies, such as differential reinforcement of alternative behaviour, that have proven effective in replacing SIB with functional, socially appropriate responses [86,187,190]. Cognitive-behavioural therapy and dialectical behaviour therapy are widely used for individuals with comorbid emotional dysregulation, particularly in adolescents and adults; however, the use is restricted to individuals with profound intellectual disabilities and those who are nonverbal [86,187,190]. These approaches enhance emotion regulation, distress tolerance, and problem-solving skills, usually combining mindfulness-based techniques with behavioural strategies to address the maladaptive coping mechanisms that perpetuate SIB [86,187,190].

As these behavioural interventions require identification of the specific reinforcement maintaining SIB, they are often based on the premise that the behaviour is maintained by socially mediated reinforcement (i.e., influenced by social interactions and/or environment). Consequently, they are more difficult to implement in the more than 25% of cases in which automatic reinforcement (i.e., sensory or alternate consequences resulting directly from the behaviour itself) of the behaviour is hypothesised [27]. Automatically reinforced SIB poses unique treatment challenges because the maintaining reinforcer is neither easily identifiable nor directly controllable by clinicians in most cases [27,193]. Recently, however, Hagopian et al. identified a predictive behaviour marker for response to reinforcement-based interventions, showing that SIB occurring primarily in an “alone” condition responds better than SIB that remains high across all conditions, marking a first step toward effective treatment stratification for automatically reinforced SIB [194].

### 5.2. Pharmacological Interventions

When behavioural interventions alone are insufficient to manage SIB, pharmacologic therapy may be considered an additional option; however, pharmacotherapy is most effective when combined with behavioural interventions, as medications primarily address symptoms rather than the underlying mechanisms [65]. Pharmacological treatments for SIB often target neurotransmitter systems implicated in its pathophysiology, such as dopaminergic and serotonergic pathways, among others [189,191,192].

Second-generation antipsychotics such as risperidone and aripiprazole are among the most commonly prescribed agents for managing SIB [65,86,190,195] and are FDA-approved for the symptomatic management of aggression, self-injury, and temper tantrums in children and adolescents with ASD [196]. These drugs primarily modulate dopamine (D2 receptor) and serotonin (5-HT2A receptor) activity, restoring balance in cortico-limbic-striatal circuits [65]. For instance, risperidone has significantly reduced irritability and aggression, with an 8-week, placebo-controlled trial reporting a 56.9% reduction in symptoms compared to 14.1% in the placebo group [196,197]. Similarly, aripiprazole has shown robust efficacy in reducing irritability and associated SIB in randomised controlled trials [198]. Other second-generation antipsychotics, such as olanzapine, paliperidone, and ziprasidone, had shown positive results for managing other challenging behaviours (e.g., irritability) in children with ASD [65].

Other pharmacological agents include SSRIs, such as fluoxetine, which enhance serotonergic tone and have been shown to reduce impulsivity and repetitive behaviours [189,191,192]. Additionally, N-acetylcysteine, a glutamatergic modulator with antioxidant properties, has shown promise in reducing repetitive and injurious behaviours by restoring excitatory-inhibitory balance within cortico-striatal circuits [189,191,192].

Opioid antagonists, particularly naltrexone and naloxone, have been explored as potential treatments for self-injurious behaviour based on the hypothesis that dysfunctions in the endogenous opioid system contribute to the maintenance of SIB in some individuals [199,200]. Naltrexone, an orally administered opioid antagonist, has shown promise in reducing self-injury, with a quantitative review of 27 studies reporting that 80% of individuals demonstrated improvement relative to baseline, and 47% exhibited a reduction in SIB of 50% or more [199,200]. While the precise mechanism remains unclear, it has been suggested that opioid antagonists may reduce SIB either by blocking the euphoric or analgesic effects associated with self-injury or through more general anxiolytic and sedative properties. Naltrexone’s efficacy appears to vary among individuals, with some studies indicating a dose-dependent effect and a higher likelihood of response in males [199,200]. Despite positive findings, methodological limitations, including small sample sizes and variability in dosing, necessitate further controlled studies to determine optimal treatment protocols.

Cannabidiol (CBD) has been investigated as a potential treatment for SIB in ASD based on the influence of the endocannabinoid system on emotional regulation and behavioural control [201]. Some studies suggest that CBD extracts may reduce self-injury, aggression, and irritability in children and adolescents with ASD; however, methodological inconsistencies, including variations in CBD formulations, dosing, and outcome measures, limit the reliability of findings and lower the quality of evidence [201]. Despite growing interest in CBD for managing behavioural symptoms in ASD, robust, well-controlled randomised controlled trials are needed to determine its true efficacy and safety [201].

Most pharmacologic treatments are prescribed off-label, relying on clinical judgement tailored to each case, with careful, ongoing monitoring of therapeutic effects [65]. Given that the precise pathophysiology of SIB remains unclear, current pharmacologic treatments focus mainly on alleviating symptoms rather than targeting the underlying mechanistic causes [86,187,190]. Despite their efficacy, pharmacological treatments are associated with potential side effects, including weight gain, sedation, and extrapyramidal symptoms, necessitating careful monitoring and individualised treatment planning [86,187,190].

### 5.3. Neuromodulation Therapies

For individuals with severe, treatment-refractory SIB, neuromodulatory approaches, such as deep brain stimulation (DBS), may present novel therapeutic options for this population when other treatments are not tolerated or effective [68,202,203]. This type of intervention, however, is still experimental and reserved for highly refractory cases where there is a foreseeable risk of injury. DBS involves delivering targeted electrical stimulation to key nodes within the fronto-limbic-striatal network, and various targets, including the amygdala, posterior hypothalamus, and nAcc, have been studied [11,70,82,202,203,204,205]. As described above, these regions are implicated in the regulation of aggression, impulsivity, and repetitive behaviours, all of which are disrupted in patients with SIB. Reports show that high-frequency stimulation of these targets and networks may improve SIB symptomology [68,202,203]. The stimulation of spatially distant targets may modulate a convergent common network of distant brain areas, including the amygdala, insula, and anterior cingulate [70].

The basolateral amygdala (BLA) has received increasing attention as a DBS target due to its critical role in regulating emotional processing, aggression, and fear responses [206]. In a pioneering case report, Sturm et al. successfully applied DBS in the BLA of a teenage patient with severe Kanner’s autism and life-threatening SIB [207]. The patient, who had previously been unresponsive to pharmacological and behavioural interventions, exhibited a substantial reduction in SIB severity and broader improvements in core symptoms of ASD [207]. The most significant improvements were observed when stimulation was restricted to the BLA, with no beneficial effects noted from stimulation of adjacent regions, such as the central amygdala or supra-amygdaloid projection system [207]. The observed improvements align with the hypothesised role of the BLA as a central hub for integrating sensory and emotional information, suggesting that dysregulation of BLA circuits contributes to the pathogenesis of SIB and ASD-related symptoms [207].

Mechanistically, BLA-DBS is thought to restore balance within excitatory and inhibitory circuits, particularly through modulation of GABAergic and glutamatergic signalling [207,208,209]. Hyperexcitable states within the amygdala, arising from reduced GABAergic inhibition, have been linked to heightened emotional reactivity and maladaptive behaviours [207,208,209]. By interfering with these hyperactive networks, BLA-DBS may suppress SIB’s emotional and impulsive drivers. Additionally, the therapeutic effects of BLA-DBS may involve broader network-level changes, as the BLA is highly interconnected with regions such as the PFC, orbitofrontal cortex, and striatum [206].

The posterior hypothalamus (pHyp) has emerged as a prominent target for DBS in treatment-refractory SIB and aggression, owing to its central role in autonomic regulation and its involvement in aggression-related circuits [11,82,203,204,205,210,211,212,213]. Clinical studies have demonstrated substantial reductions in the frequency and severity of aggressive behaviours following pHyp-DBS. In a recent multicenter analysis, we reported symptom improvement in 91% of patients with severe refractory aggression, with reductions ranging from 38% to 100% [203]. In addition to improved behavioural outcomes, patients exhibited reduced agitation and significantly enhanced overall quality of life [11,82,203,204,205,210,211,212,213]. Notably, in patients with comorbid epilepsy, a marked decrease in seizure frequency is also observed following pHyp-DBS, indicating broader therapeutic effects [203,204,210].

Functional connectivity analyses identified strong links between the pHyp and regions critical to aggression regulation, particularly the periaqueductal grey and key limbic structures, as predictors of treatment response [203]. Additionally, connectivity with brainstem nuclei responsible for monoamine synthesis—including the dorsal raphe nuclei (5-HT), substantia nigra and ventral tegmental area (DA), and the locus coeruleus (norepinephrine)—was strongly associated with symptom improvement [203,204]. These findings underscore the role of the posterior hypothalamus as a critical hub within a broader aggression-modulating network, mediating its therapeutic effects through extensive connections with the limbic system and brainstem monoaminergic pathways.

The nAcc, a key structure within the ventral striatum, plays a central role in reward processing, impulsivity, and motor control [68,70,202,214]. Its involvement in integrating dopaminergic reinforcement signals within the mesolimbic reward pathway makes it a critical target for DBS in neuropsychiatric conditions marked by impaired inhibitory control and by dysfunction in frontostriatal dynamics, such as OCD, addiction, and mood disorders [214,215,216,217,218]. In the context of SIB, we recently reported the results of phase I clinical trial investigating nAcc-DBS in a cohort of paediatric patients, showing that stimulation of the nAcc led to significant reductions in SIB severity and associated repetitive behaviours, with concurrent improvements in quality of life and decreased reliance on physical restraints [202]. Neuroimaging evaluation following nAcc-DBS revealed decreased abnormal metabolic activity in the thalamus, striatum, and temporoinsular cortex [202]. This indicates that the therapeutic effects may stem from normalising hyperactive cortico-striatal circuits implicated in SIB [202]. These findings highlight the nAcc as a promising neuromodulatory target for SIB, particularly in cases refractory to conventional treatments. By influencing dopamine-mediated reward pathways and regulating frontostriatal activity, nAcc-DBS has the potential to address the impulsive and repetitive components of SIB while improving overall behavioural regulation.

While DBS of key nodes along the fronto-limbic-striatal network has shown significant promise for managing severe, treatment-resistant SIB, several challenges remain. Optimal stimulation parameters, target selection, and patient-specific predictors of treatment response require further investigation through larger, well-controlled clinical trials. Additionally, long-term studies are needed to assess the durability of DBS effects and potential adverse outcomes, particularly in paediatric populations.

### 5.4. Molecular-Targeted Therapies

Recent advances in molecular biology have identified potential targets for novel therapies. Dysregulation in glutamatergic signalling within cortico-striatal circuits has been addressed using metabotropic glutamate receptor 5 (mGluR5) antagonists, which reduce repetitive and injurious behaviours in preclinical models [124,219,220]. These findings suggest modulating glutamatergic excitability could mitigate SIB across neurological, neurodevelopmental, and neuropsychiatric conditions. Other promising targets include neuropeptides, such as oxytocin, which have garnered interest for their role in modulating social behaviour and emotional regulation [221]. Preclinical studies suggest that oxytocin administration may reduce SIB by enhancing prosocial interactions and improving stress reactivity. In patients with ASD, oxytocin administration led to improvement in eye contact and emotion recognition measures; however, chronic use was associated with the development of side effects [222,223].

Adenosine receptor signalling, particularly involving A2A receptors, has emerged as a promising target for mitigating repetitive and SIB by restoring striatal function and regulating excitatory-inhibitory balance [224,225]. The adenosinergic system is critical in modulating neurotransmission, interacting closely with DA and glutamate systems, particularly within cortico-striatal circuits implicated in motor control and behavioural regulation [224]. Dysfunction in the adenosine-DA interplay has been linked to hyperactivity within the direct pathway and impaired inhibitory control [95,224,225]. Pharmacological studies have demonstrated that A2A receptor agonists can effectively reduce repetitive behaviours by modulating the overactive striatal circuits and restoring the balance between excitatory glutamatergic and inhibitory GABAergic signalling [95,224,225]. This effect has been particularly notable when combined with A1 receptor modulation, further enhancing the inhibitory tone and normalising cortico-striatal dynamics [95,224,225].

The management of SIB requires a multifaceted approach that integrates psychosocial interventions, pharmacotherapy, neuromodulation, and emerging molecular-targeted therapies. While behavioural strategies remain the foundation of treatment, advances in understanding SIB’s neurobiological and molecular basis provide exciting opportunities for targeted, precision-based interventions. Future research should prioritise large-scale clinical trials and translational studies to optimise treatment efficacy and address the unique needs of individuals affected by this debilitating behaviour.

## 6. Conclusions and Future Directions

SIB represents a severe form of aggression that profoundly impacts patients, caregivers, and healthcare systems [226,227]. Its high prevalence across neurodevelopmental, neuropsychiatric, and genetic disorders underscores the urgent need for a deeper understanding of its underlying mechanisms to inform more effective therapeutic strategies [13,14]. This review has synthesized evidence from clinical, preclinical, and translational studies to provide a comprehensive overview of the neurobiological basis, etiological factors, animal models, and treatment approaches for SIB.

The genetic and epigenetic contributions to SIB highlight the interplay between inherited susceptibility, molecular alterations, and environmental influences [18,19]. Genetic mutations and polymorphisms in key pathways and epigenetic modifications provide critical links between environmental stressors and gene expression changes that exacerbate vulnerability to SIB [18,19]. Understanding these gene-environment interactions is essential for identifying specific biomarkers and developing novel precision-based therapeutic strategies targeting these molecular disruptions. Future research should prioritize multiomic approaches, including genomic, epigenomic, and transcriptomic studies, to better elucidate the biological underpinnings of SIB.

The neurobiology of SIB reflects a convergence of neurotransmitter dysregulation, notably within dopaminergic, serotonergic, glutamatergic, and GABAergic systems, and structural and functional abnormalities in fronto-limbic-striatal circuits [17,68,69,70]. These findings are supported by insights gained from both human studies and validated animal models, which highlight the interaction between genetic predispositions, epigenetic regulation, and environmental stressors in driving this complex behaviour [29,108]. Continued use of animal models, coupled with innovative techniques such as optogenetics, chemogenetics, and in vivo imaging, will enable precise dissection of circuit-level dynamics and molecular targets underlying SIB. This work remains critical for bridging the gap between preclinical findings and clinical applications.

In terms of treatment, a multifaceted approach integrating psychosocial, pharmacological, and neuromodulatory interventions is essential for addressing the complexity of SIB [68,86,187,188]. Behavioural therapies remain the gold standard for managing SIB, particularly when tailored to individual triggers and environmental contexts [68,86,187,188]. Pharmacological treatments provide symptomatic relief but require careful monitoring for side effects, especially in paediatric populations [189,191,192]. Emerging molecular-targeted therapies promise to restore excitatory-inhibitory balance and reduce repetitive behaviours [220,224]. Neuromodulation techniques targeting key nodes within the fronto-limbic-striatal network have resulted in substantial reductions in SIB severity, improved quality of life, and enhanced functional outcomes [11,70,82,202,203,204,205]. Future research should focus on optimising stimulation parameters, refining patient selection criteria, and elucidating the long-term effects of neuromodulation, particularly in vulnerable populations such as children and individuals with severe intellectual disabilities.

Looking ahead, translational research that integrates molecular biology, neuroimaging, and circuit-level analyses will be pivotal for developing precision-based interventions. Large-scale, longitudinal studies are needed to identify reliable biomarkers for SIB, improve early detection, and evaluate the efficacy of emerging therapies in diverse patient populations. Furthermore, a systems-level approach that considers the interaction between genetic, environmental, and neurobiological factors will deepen our understanding of shared mechanisms across neuropsychiatric conditions.

In conclusion, while significant strides have been made in understanding the neurobiology and treatment of SIB, substantial gaps remain. By leveraging advances in preclinical models, molecular-targeted therapies, and neuromodulation techniques, future research can transform the clinical management of SIB. A multidisciplinary, personalised approach that targets the root causes of SIB, rather than simply alleviating its symptoms, promises to improve outcomes for individuals affected by this debilitating behaviour.

## Figures and Tables

**Figure 1 ijms-26-01938-f001:**
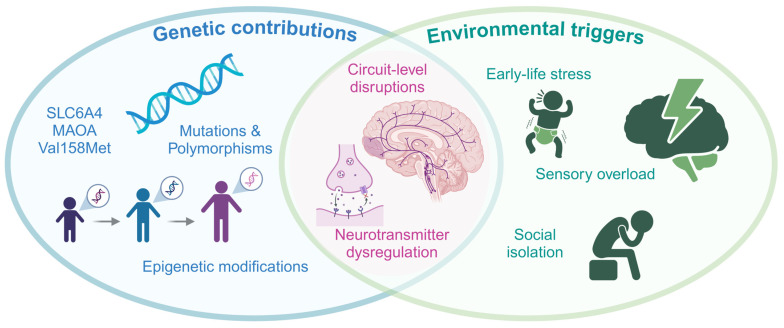
Diagram illustrating how genetic predispositions and environmental factors interact to influence SIB expression.

**Figure 2 ijms-26-01938-f002:**
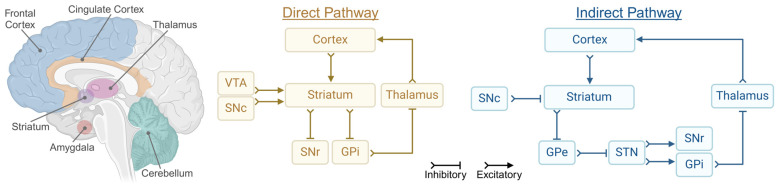
Neurocircuitries regulating self-injury behaviour. Abbreviations: GPe: Globus pallidus externus; GPi: Globus pallidus internus; SNc: Substantia nigra pars compacta; SNr: Substantia nigra pars reticulata; STN: Subthalamic nucleus; VTA: Ventral-tegmental area.

**Table 1 ijms-26-01938-t001:** Summary of animal models of self-injurious behaviour and key neurobiological findings.

Category	Model	Details
Genetic	*Shank3^−/−^*	Elevated SIB and skin lesions related to reduced Shank3 expression in the hippocampus and cortex [48] Increased repetitive behaviour associated with reduced Shank3 in the striatum [48] Reduced levels of SAPAP3, Homer-1b/c, PSD93, and glutamate receptor subunits GluR2, NR2A, and NR2B in striatum [49] Increased medium spiny neuron complexity [49]
	BTBR	Reduced DA levels and increased DA metabolites in amygdala [50] Elevated glutamate levels in amygdala [50] Imbalance of GABA and glutamate levels in amygdala, prefrontal cortex, and hippocampus [51] Reduced SERT and increased 5-HT1A receptor signalling capacity [52,53] Decreased striatum and thalamus volume and increased hippocampus, cerebral cortex, and cerebellum volume correlated with elevated self-grooming [34]
	*Fmr1^−/−^*	Deficits in striatal GABA, glutamate, and 5-HT associated with hyperactivity, social and cognitive impairment [54] GABA_A_ receptor agonist (gaboxadol) normalised aberrant behaviours [55] Abnormal dendritic spine density and phenotypes [56,57]
	*Mecp2* mutation	Mecp2^−/y^mice exhibit reduced responses of parvalbumin-expressing inhibitory neurons and altered polarity of GABAergic inhibition in pyramidal neurons [58]
	*Hoxb8^−/−^*	Increased cortical synapse and spine density in the frontal cortex; increased dendritic spines in dorsal- and ventromedial striatum [59] Mutants transplanted with healthy bone marrow (i.e., normal microglia) and reduced excessive pathological grooming [60] SIB induced by optogenetic stimulation of Hoxb8 microglia in dorsomedial striatum or medial prefrontal cortex [61]
	*Slc6a3^−/−^*	Elevated levels of extracellular DA associated with hyperactivity, impulsivity, repetitive behaviour, and SIB [62]
Lesion models	Neonatal 6-OHDA lesion	D1/5 agonist (SKF 38393) reduces SIB and inhibits responsiveness of spontaneously firing striatal units [63] Higher levels of striatal GABA, met-enkephalin, and substance P in adulthood [64,65] Hyperinnervation of striatal 5-HT neurons in adulthood, accompanied by increased 5-HT_1B_ and 5-HT_2_ receptor binding and supersensitivity to 5-HT receptor antagonists [66,67]
Early environmental deprivation	Social deprivation	Diazepam decreases self-wounding episodes in captive rhesus macaques [68] Hypercortisolemia exhibited by individually caged rhesus macaques with SIB [69] Elevated striatal DA levels in isolation-reared rats [70] Maternally deprived macaques have lower concentrations of DA metabolite, DOPAC, in cerebrospinal fluid [71] Isolation-reared rhesus monkeys exhibit increased stereotyped behaviours after apomorphine administration when compared to group-housed animals [72]
	Environmental stress	Footshock stress increases SIB and DA concentrations in the striatum and frontal cortex of neonatal 6-OHDA lesioned rats [73,74,75] Nesting material enrichment reduces SIB in individually housed rat models [76] Acute stress altered the functioning of the LHPA axis, including blunted cortisol response [77,78]
Pharmacologic	Psychostimulant (Pemoline, Methamphetamine, Amphetamine)	Pemoline induces SIB within 48 hrs of a single 250–300 mg/kg dose or after 3–12 daily injections of 80–200 mg/kg/day [79] Pemoline-induced SIB reduced by DA and 5-HT receptor antagonists (haloperidol, pimozide, and risperidone) [80], and NMDA receptor antagonist (MK-801) [81] Pemoline-induced SIB enhanced by paroxetine (an SSRI) [82] DA injection increased evoked depolarising potential responses of neurons in rats with pemoline-induced SIB [83] SCH23390 (D1R antagonist) and 5-HTP (5-HT metabolic intermediate) administration reduced methamphetamine-induced SIB; no effect after sulpiride or naloxone injection [84] Methamphetamine-induced SIB reduced after MK-801 administration [85] Dose-dependent increase in stereotypic behaviour, oral dyskinesia, and SIB in response to amphetamine treatment [86] Risperidone decreased amphetamine-induced SIB; haloperidol and SCH23390 were ineffective [86]
	Bay K 8644	Effects are blocked by L-type calcium channel antagonists [87,88] SIB reduced by monoamine oxidase inhibitor [89], indirect DA agonists [90], D_1/5_ and D_3_ antagonists [91] SIB unaffected by D_2_ and D_4_ antagonists [91] SIB enhanced by fluoxetine (an SSRI) and decreased by 5-HT depletion [89]
	Clonidine	Mixed reports for alpha-adrenoreceptor antagonist effects on clonidine-induced SIB [92,93]
	Chronic caffeine	A small percentage exhibit SIB, with minor frequency and severity High doses are toxic (i.e., weight loss, thymus involution, chromodacryorrhea, death) [83,94]

Abbreviations: 5-HT, serotonin; 6-OHDA, 6-hydroxydopamine; DA, dopamine; DOPAC, 3,4-dihydroxyphenylacetic acid; GABA, gamma-aminobutyric acid; LHPA, limbic-hypothalamic-pituitary-adrenal; NMDA, N-methyl-D-aspartate; SERT, serotonin transporter; SIB, self-injurious behaviour; SSRI, serotonin reuptake inhibitor.

**Table 2 ijms-26-01938-t002:** Summary of main therapeutics used for the clinical management of self-injurious behaviours.

Category	Therapy	Biological Basis	Specific Disorders Studied	Available Data
Pharmacologic	Risperidone	5-HT receptor agonist D2 receptor agonist Adrenoreceptor agonist	ASD; PDD; Down syndrome; FXS; Schizophrenia; BP; OCD; ADHD; MDD	Randomized, placebo-controlled trials Open-label extension trial
	Aripiprazole	Partial D2 receptor agonist 5-HT receptor agonist	ASD	Case series Randomized, placebo-controlled trials
	Clonidine	Adrenoreceptor agonist Inhibits excitatory cardiovascular neurons Reduces sympathetic outflow	PDD; ADHD	Case series Open-label pilot study
	N-Acetylcysteine	Restores glutathione Scavenges oxidants	ASD	Case report Randomized, double blind, placebo controlled studies
	Riluzole	Inhibits glutamate release and enhances glutamate reuptake Inactivates voltage dependent Na+ channels	FXS; ASD	Case series Randomized, double blind, placebo controlled trial
	Mirtazapine	Adrenergic antagonist 5-HT receptor antagonist H1 receptor antagonist	PDD; ASD	Open-label study
	Naltrexone	Opioid antagonist	PWS	Case reports Case series
	Topiramate	Blocks neuronal voltage gated Na+ channels Enhances GABA activity Glutamate receptor antagonist Carbonic anhydrase inhibitor	PWS; ASD	Case series Open-label trial Double blind, placebo controlled trial
Behavioural	Behavioural therapy (multiple)	Target maladaptive thoughts and behaviours Identifies proximal stressors	ASD; BPD; Eating disorder; TS; OCD	Case series Case reports Randomized controlled trials
Neuromodulation	ECT	Induces controlled seizures, which alter the chemical and electrical architecture of the brain Disrupts neural circuits, affecting neuroplasticity Alters neurotransmitter systems	ASD; MDD; Catatonia; OCD; BD; Schizophrenia	Case series Case reports
	Deep brain stimulation	Produces electrical impulses that affect brain activity (i.e., oscillatory activity, neurochemistry, plasticity, etc.)	TS; OCD; ASD; Dyskinesia; Acquired brain injury; Epilepsy; Dystonia	Case series Case reports Open-label pilot study

Abbreviations: 5-HT, serotonin; ADHD, attention deficit hyperactivity disorder; ASD, autism spectrum disorder; BP: bipolar disorder; BPD: borderline personality disorder; D2, dopamine receptor 2; ECT: electroconvulsive therapy; FXS: Fragile X syndrome; GABA, gamma-aminobutyric acid; MDD: major depression disorder; OCD, obsessive-compulsive disorder; PDD: pervasive developmental disorder; PWS: Prader-Willi syndrome; TS: Tourette syndrome.

## Data Availability

No new data were created or analysed in this study.

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
