# Peer review of "Molecular Pathways, Neural Circuits and Emerging Therapies for Self-Injurious Behaviour"

_ijms, 2025, doi:10.3390/ijms26051938_

Round 1
Reviewer 1 Report
Comments and Suggestions for Authors
This article is well-structured and thoroughly explores the various aspects of SIB, from potential mechanisms to new therapies.
I would find it interesting and usefull to delve deeper into the difference between socially maintained SIB and automatically reinforced SIB in the various areas discussed in the paper, especially regarding the therapeutic aspects.
Author Response
Author’s Response: We thank the reviewer for raising this important point. In addition to describing socially maintained and automatically reinforced forms of SIB in the Introduction, paragraph 4, we have added a paragraph in Section 5.1. addressing this point:
5.1. Behavioural Interventions, Paragraph 3:
“As these behavioural interventions require identification of the specific reinforcement maintaining SIB, they are often based on the premise that the behavior is maintained by socially mediated reinforcement (ie. influenced by social interactions and/or environment). Consequently, they are more difficult to implement in the more than 25% of cases in which automatic reinforcement (ie. sensory or alternate consequences resulting directly from the behavior itself) of the behavior is hypothesized [27]. Automatically reinforced SIB poses unique treatment challenges because the maintaining reinforcer is neither easily identifiable nor directly controllable by clinicians in most cases [27,193]. Recently however, Hagopian et al. identified a predictive behaviour marker for response to reinforcement-based interventions, showing that SIB occurring primarily in an “alone” condition responds better than SIB that remains high across all conditions, marking a first step toward effective treatment stratification for automatically reinforced SIB [194].”

Reviewer 2 Report
Comments and Suggestions for Authors
The authors presented an excellent review manuscript entitled “Molecular Pathways, Neural Circuits and Emerging Therapies for Self-Injurious Behaviour”. This is an outstanding review article, certainly with few manuscripts with similar high-quality pattern in the current literature. Figures 1 and 2 summarize nicely several aspects described in the text. Table 2 presents also an interesting summary of different therapies involved in the treatment of self-injurious behaviour. A minor suggestion for authors is to review the genes descriptions in the text putting all of them in italics (presented in items 2.1, 2.2, 2.3). Furthermore, the clinical correlations presented in item 2.1 are also of interest, however relatively few conditions were described – I suggest the inclusion of some of the disorders presented, for example, in OMIM Database (URL: https://www.omim.org/search?index=entry&start=1&limit=10&sort=score+desc%2C+prefix_sort+desc&search=self-injury).
Author Response
Author’s Response: We thank the reviewer for the suggestions. All gene names have been italicized where necessary throughout the manuscript. We also added descriptions of additional genetic conditions associated with SIB, namely Cri-du-Chat, Angelman, Lowe and Cornelia de Lange syndromes. Due to the vastly heterogeneous genetic landscape of SIB and its related disorders, we chose to describe the genetic syndromes that have been more widely studied in the context of aggressive behaviours and SIB. These additional disorders are incorporated in Section 2.1.
